# Cerebrospinal fluid endo-lysosomal proteins as potential biomarkers for Huntington's disease

**Alexander J. Lowe**[1], **Simon Sjödin**[2], **Filipe B. Rodrigues**[1], **Lauren M. Byrne**[1], **Kaj Blennow**[2,3], **Rosanna Tortelli**[1], **Henrik Zetterberg**[1,2,3,4], **Edward J. Wild**[1] *

**1** UCL Huntington's Disease Centre, UCL Queen Square Institute of Neurology, University College London, London, United Kingdom, **2** Department of Psychiatry and Neurochemistry, Institute of Neuroscience and Physiology, The Sahlgrenska Academy at the University of Gothenburg, Mölndal, Sweden, **3** Clinical Neurochemistry Laboratory, Sahlgrenska University Hospital, Mölndal, Sweden, **4** UK Dementia Research Institute at UCL, London, United Kingdom

* e.wild@ucl.ac.uk

## Abstract

Molecular markers derived from cerebrospinal fluid (CSF) represent an accessible means of exploring the pathobiology of Huntington's disease (HD) in vivo. The endo-lysosomal/autophagy system is dysfunctional in HD, potentially contributing to disease pathogenesis and representing a potential target for therapeutic intervention. Several endo-lysosomal proteins have shown promise as biomarkers in other neurodegenerative diseases; however, they have yet to be fully explored in HD. We performed parallel reaction monitoring mass spectrometry analysis (PRM-MS) of multiple endo-lysosomal proteins in the CSF of 60 HD mutation carriers and 20 healthy controls. Using generalised linear models controlling for age and CAG, none of the 18 proteins measured displayed significant differences in concentration between HD patients and controls. This was affirmed by principal component analysis, in which no significant difference across disease stage was found in any of the three components representing lysosomal hydrolases, binding/transfer proteins and innate immune system/peripheral proteins. However, several proteins were associated with measures of disease severity and cognition: most notably amyloid precursor protein, which displayed strong correlations with composite Unified Huntington's Disease Rating Scale, UHDRS Total Functional Capacity, UHDRS Total Motor Score, Symbol Digit Modalities Test and Stroop Word Reading. We conclude that although endo-lysosomal proteins are unlikely to have value as disease state CSF biomarkers for Huntington's disease, several proteins demonstrate associations with clinical severity, thus warranting further, targeted exploration and validation in larger, longitudinal samples.

## Introduction

Huntington's disease (HD) is an autosomal dominant, neurodegenerative disease characterised by progressive motor, psychiatric and cognitive dysfunction [1]. An extended polyglutamine

**Data Availability Statement:** All relevant data are within the paper and its Supporting Information files.

**Funding:** This work was supported in part by the National Institute for Health Research University College London Hospitals Biomedical Research Centre, the UCL Leonard Wolfson Experimental Neurology Centre, and the Swedish Research Council. E.J.W. has research funding from the Medical Research Council UK (MR/M008592/1) (https://mrc.ukri.org/funding/), CHDI Foundation Inc (https://chdifoundation.org/) and European Huntington's Disease Network (http://www.ehdn.org/). HZ is a Wallenberg Scholar supported by grants from the Swedish Research Council (#2018-02532) (https://www.vr.se/english.html), the European Research Council (#681712) (https://erc.europa.eu/), the Swedish state under the agreement between the Swedish government and the County Councils, the ALF-agreement (#ALFGBG-720931), the Alzheimer Drug Discovery Foundation, USA (#201809-2016862) (https://www.alzdiscovery.org/research-and-grants/), and the UK Dementia Research Institute at UCL (https://www.ucl.ac.uk/uk-dementia-research-institute/). The funders had no role in study design, data collection and analysis, decision to publish, or preparation of the manuscript.

**Competing interests:** I have read the journal's policy and the authors of this manuscript have the following competing interests: AJL, FBR, LMB, RT, HZ and EJW are University College London employees. FBR has provided consultancy services to GLG and F. Hoffmann-La Roche Ltd. EJW reports grants from Medical Research Council, CHDI Foundation, and F. Hoffmann-La Roche Ltd during the conduct of the study; personal fees from Hoffman La Roche Ltd, Triplet Therapeutics, PTC Therapeutics, Shire Therapeutics, Wave Life Sciences, Mitoconix, Takeda, Loqus23. All honoraria for these consultancies were paid through the offices of UCL Consultants Ltd., a wholly owned subsidiary of University College London. HZ has served at scientific advisory boards for Denali, Roche Diagnostics, Wave, Samumed and CogRx, has given lectures in symposia sponsored by Fujirebio, Alzecure and Biogen, and is a co-founder of Brain Biomarker Solutions in Gothenburg AB (BBS), which is a part of the GU Ventures Incubator Program. This does not alter our adherence to PLOS ONE policies on sharing data and materials.

tract (polyQ) in the ubiquitously-expressed Huntingtin protein (HTT), results in the production of a mutated, pathogenic product (mHTT) which accumulates intracellularly causing toxicity and neuronal death [2, 3].

Neuronal survival is dependent, among other things, on intracellular surveillance mechanisms including autophagy, a lysosomal pathway that serves to eliminate toxic substances via three mechanisms, namely microautophagy, macroautophagy and chaperone-mediated autophagy (CMA) [4]. Macroautophagy involves the engulfment of cargo into a double-membrane sequestering vesicle known as an autophagosome. Following fusion with vesicles from the endo-lysosomal compartment, an autolysosome is formed in which the cargo is degraded by lysosomal hydrolases and the resultant macromolecules are released back into the cytosol [5]. Microautophagy and CMA do not involve the formation of an autophagosome, instead using direct import or intraluminal vesicle formation to engulf cargo into the endo-lysosomal compartment [6]. Despite their differences, mechanistic crossovers between the three autophagy pathways have been described. Lysosomal-associated membrane protein-2 splice variant A (LAMP-2A), previously only described in CMA, has been shown to be important for syntaxin-17 mediated vesicle fusion in macroautophagy [7]. Furthermore, syntaxin-17 is pivotal for targeting mitochondrial-derived vesicles to the endo-lysosomal compartment for degradation in microautophagy [8].

Macroautophagy plays a pivotal role in the clearance of aggregated proteins via aggrephagy [4, 9], whereby aggregates are selectively bound to the autophagosome membrane through the action of adaptor proteins, including p62 and Alfy [10]. Autophagy disruptions have been reported in several neurodegenerative diseases including HD, in which basal autophagy appears to function normally; however, the autophagosomes are devoid of cargo, as recruitment of mHTT to the organelle fails [11–17]. Interestingly, HTT shows structural similarities to three selective autophagy proteins in yeast [18, 19] and promotes selective macroautophagy in mammalian cells by mediating the binding of p62 and the autophagy-initiating kinase, UKL1 [20]. As such, it is possible that the polyglutamine expansion in HD may disrupt HTT's role in selective autophagy [21]; however, studies have shown that autophagic clearance of aggregates can still occur despite overexpression of mHTT in mice and cellular models [22, 23]. In light of HTT's role as an autophagic scaffold protein, the mechanistic crossovers between the three pathways, and their possible contribution to neurodegeneration, we sought to study the alterations and autophagic dysfunction in HD mutation carriers and controls.

Lysosomal-associated membrane protein-2 (LAMP2) has pivotal roles in autophagy including translocation of cargo into the lumen and as a receptor in CMA [24, 25]. LAMP2 gene expression levels and total levels of LAMP2 protein have been shown to be reduced and increased in PD and AD respectively [26–29]. Additionally, cerebrospinal fluid (CSF) LAMP2 has been indicated as a potential biomarker in AD with increased concentration compared to controls [28, 30] and has been found to correlate with phosphorylated tau, a well-established marker of neuronal pathology [31]. In HD, a compensatory increase in CMA has been described in response to defective macroautophagy which may explain the increased mRNA expression of LAMP2 and increased levels of LAMP2 protein in HD cell models [16].

Deficits in lipid synthesis and metabolism, both of which are reported in HD [32], could contribute towards autophagy failure [33]. Glycosphingolipids endocytosed from the plasma membrane are degraded in the lysosome via the synchronous activity of hydrolases and activator proteins [34]. Ganglioside GM2 activator (GM2A) is a lysosomal protein that together with beta-hexosaminidase-β (HEXB), catalyses the degradation of gangliosides, specifically GM2 [35]. GM2A has shown promise as a CSF biomarker for neurodegeneration in AD, correlating with CSF amyloid-beta levels, and in Lewy body dementia (LBD) with increased concentration [36], whilst the concentration in PD has shown to be reduced [37]. The reason for elevated

CSF GM2A in AD and LBD is currently unknown but likely reflects generalised lysosomal dysfunction, as elevated GM2A has been detected via urinary analysis in lysosomal storage disorders [38]. In HD, the reduced expression of genes involved in ganglioside catabolism has been reported [39], in addition to disturbances in ganglioside metabolism and synthesis [39, 40]. Furthermore, administration of gangliosides has been found to reduce apoptosis in HD cell lines and restore normal ganglioside concentration in YAK128 mice, resulting in improved motor function [40, 41]. Given that gangliosides are involved in regulating white matter integrity [42], and that white matter atrophy is associated with HD [43–45], the exploration of CSF GM2A, a protein pivotal for ganglioside catabolism, is warranted and may further explain white matter pathology in HD.

Lysosomal proteolytic degradation involves the activity of the cathepsin family of proteases [46]. Previous work using CSF has demonstrated significant alterations in the concentration of several cathepsins in other proteopathies such as PD [37]. Both Cathepsin L and Z have been shown to be crucial for the degradation of polyQ proteins within lysosomes [47], suggesting a protective role against toxic aggregates. The role of additional cathepsins in HD has also been explored, with early work describing an increase in Cathepsin D activity in caudate tissue of HD patients [48]. This has been supported by recent studies showing increased Cathepsin D and L levels in response to mHTT expression in vitro [49], and studies demonstrating overexpression of Cathepsin B and D to reduce mHTT levels and toxicity in multiple cell models, without impacting upon endogenous HTT [50].

CSF is enriched in brain-derived substances, thus biomarkers derived from CSF represent a valid means to assess neuropathology [51]. Given the dysregulation of the autophagy pathway in HD [33], the exploration of endo-lysosomal proteins in HD patients could represent a means of identifying novel biomarkers with prognostic, disease monitoring and pharmacodynamic value [52]. Parallel reaction monitoring mass spectrometry (PRM-MS) is a quantitative approach making use of high resolution instruments and thus offers highly selective and accurate measurements [53, 54]. Separation in two dimensions, by physiochemical properties using liquid chromatography and by mass to charge ratio (m/z) using mass spectrometry, facilitates multiplexing capabilities in complex matrices, for example in biofluids. The PRM-MS method employed herein has previously been applied to investigate endo-lysosomal dysfunction in AD and PD patients, with the later demonstrating altered CSF concentrations of multiple cathepsins, GM2A and LAMP2 [37].

We employed PRM-MS to conduct a targeted analysis of 18 proteins associated with endocytosis and lysosomal function in the CSF from the HD-CSF cohort baseline (60 HD mutation carriers and 20 healthy controls). Given the previously described autophagic dysfunction in HD, and their role in other neurodegenerative diseases, we pre-specified 5 lysosomal proteins as primary analytes to study: LAMP1, LAMP2, GM2A, Cathepsin D and F. The remaining 13 proteins, pertaining to other aspects of the endo-lysosomal and ubiquitin-proteasome system, were assessed in a separate exploratory analysis. We aimed to elucidate the biomarker potential of endo-lysosomal proteins whilst also highlighting targets for future comprehensive analysis, with the aim of facilitating therapeutic developments in HD.

## Materials and methods

### Participants and study design

HD-CSF was a prospective single-site study with standardised longitudinal collection of CSF, blood and phenotypic data (online protocol: 10.5522/04/11828448.v1). Ethical approval was given by the London Camberwell St Giles Research Ethics Committee, with all participants providing written informed consent prior to enrolment. The study involved manifest HD,

premanifest HD and healthy controls. Manifest HD was defined as UHDRS diagnostic confidence level (DCL) = 4 and CAG repeat length > 36. Premanifest HD had CAG repeat length > 40 and DCL < 4. Healthy controls were contemporaneously recruited, drawn from a population with a similar age to patients, and clinically well, so the risk of incidental neurode-generative diseases was very low. Consent, inclusion and exclusion criteria, clinical assessment, CSF collection and storage were all as previously described [55, 56]. In brief, samples were col-lected after an overnight fast at the same time of day and centrifuged and aliquoted on ice using a standardised protocol and polypropylene plasticware. Relevant aspects of clinical phe-notype were quantified using the Unified Huntington's Disease Rating Scale (UHDRS) [57]. A composite UHDRS (cUHDRS) score was generated for each subject to provide a single mea-sure of motor, cognitive and global functioning decline. This composite score, computed using four measures; Total Functional Capacity (TFC), Total Motor Score (TMS), Symbol Digit Modality Test (SDMT) and Stroop Word Reading (SWR), has been found to display the strongest relationship to HD brain pathology and enhanced sensitivity to clinical change in early HD [58]. Disease burden score (DBS) was calculated for each HD patient using the for-mula [CAG repeat length– 35.5] × age [59]. DBS estimates cumulative HD pathology exposure as a function of CAG repeat length and the time exposed to the effects of the expansion, and has been shown to predict several features of disease progression including striatal pathology [59, 60]. Baseline samples from HD-CSF have been used for this study.

## Sample preparation

Measurement of peptide concentrations was performed as previously described [37], which builds on the original method developed by Brinkmalm et al. [61]. However, some minor modifications were introduced. In short, 50 μL CSF was mixed with 50 μL of an internal stan-dard mixture containing stable isotope-labelled peptides (JPT Peptide Technologies GmbH, Berlin, Germany; Thermo Fisher Scientific Inc. Waltham, MA, USA), $^{13}$C-labelled ubiquitin (Silantes, GmbH, München, Germany) and bovine serum albumin (Sigma-Aldrich Co., Saint Louis, MO, USA), diluted in 50 mM $NH_4HCO_3$ (see S1 Table). Reduction and alkylation was performed by the addition of 50 μL 15 mM 1,4-dithiothreitol in 50 mM $NH_4HCO_3$, shaking for 30 min at + 60 ˚C, cooling down at room temperature for 30 min, and finally the addition of 25 μL 70 mM iodoacetamide in 50 mM $NH_4HCO_3$ followed by shaking at room tempera-ture in the dark for 30 min. The samples were digested by the addition of 25 μL 0.08 μg/μL sequencing grade modified trypsin (Promega Co., Madison, WI, USA) diluted in 50 mM $NH_4HCO_3$ and incubated at + 37 ˚C shaking for 18 h. Digestion was ended by the addition of 25 μL 10% trifluoroacetic acid. Solid-phase extraction was performed using Oasis® HLB 96-well μElution Plates (2 mg sorbent and 30 μm particle size; Waters Co., Milford, MA, USA) by conditioning (2x300 μL methanol), equilibration (2 × 300 μL $H_2O$), loading of samples, washing (2 × 300 μL $H_2O$), and elution (2 × 100 μL methanol). The samples were then dried by vacuum centrifugation and stored at– 80 ˚.

## Parallel reaction monitoring mass spectrometry

Prior to analysis by PRM-MS the samples were dissolved by the addition of 50 μL 50 mM $NH_4HCO_3$, and shaking at room temperature for 1 h. Forty microliters of sample were injected and separated using a Dionex™ UltiMate™ 3000 standard-LC system (Thermo Fisher Scientific Inc., Waltham, MA, USA) and a Kinetex® EVO C18 column (length 150 mm; inner diameter 2.1 mm; particle size 1.7 μm; Phenomenex Inc., Torrance, CA, USA) with a SecurityGuard™ ULTRA cartridge prefilter (Phenomenex Inc.). On a 60 minutes method, with solvents A (0.1% formic acid in $H_2O$ (v/v)) and B (84% acetonitrile and 0.1% fromic

acid in $H_2O$ (v/v)), using a flow rate of 300 μL/min, the gradient went from 3 to 5% B over one minute followed by 5 to 26% B over 48 minutes. The column temperature was set to + 50 ˚C. Separation by high-performance liquid chromatography, as described above, was performed in online mode coupled to a Q Exactive™ Hybrid Quadrupole-Orbitrap™ mass spectrometer (Thermo Fisher Scientific Inc.). Using a HESI-II ionization probe (Thermo Fisher Scientific Inc.) electrospray ionization was performed in positive ion mode with the following settings: spray voltage + 4.1 kV, heater temperature + 400 ˚C, capillary transfer tube temperature + 380 ˚C, sheath gas flow rate 25, auxiliary gas flow rate 10, and S-Lens RF level 60. Acquisition of data was performed using single microscans in parallel reaction monitoring (PRM) mode with an isolation window of *m/z* 2 centred on the second isotope of the precursor ion. The resolution setting was 70 k with an AGC target of $1 \times 10^6$ and a 256 ms injection time. Fragmentation was performed using beam-type collision-induced dissociation (higher energy collision induced dissociation [62] with optimized energies as described before [37]. The PRM method was scheduled using one-minute retention time windows. Peptide related settings are shown in S1 Table.

## Data extraction

Skyline v.19.1 [63] was used to calculate and export fragment ion peak areas. Skyline was also used to monitor and evaluate fragment ion traces and ratios, and to determine which fragment ions to include in the analysis. The ratio between tryptic peptide and isotope-labelled peptide peak area was used for quantification. In total 48 peptides from 19 proteins, including added bovine serum albumin as a control protein, were monitored. With each set of samples analysed, four quality control replicates from a CSF pool were run to normalize variation between sets of samples. In this case the samples were split in two sets, however prepared on a single occasion but analysed using PRM-MS at different points in time. The median of the first set's four quality control replicates was used for normalization by dividing the median of the second set's quality control median. Then the samples in the second set were divided by the resulting normalization quotient (one for each peptide). As multiple peptides were monitored from each protein the complexity of the data was reduced by transforming the peptide ratios into a single value, see Eq 1. The transformation was done for proteins with correlating peptides. To create a protein-level estimate, a Mean Peptide Ratio was calculated by dividing the peptide ratio (x) by the mean of all ratios for that peptide in the study (x̄). The calculation was made for peptides 1-n, and was then divided by the number of peptides (n) derived from the protein. Thus, the sample ratios for each peptide were normalized to have a mean of 1, without affecting the relative difference between samples. Additionally, the weight of each peptide in the calculation of the Mean Peptide Ratio became approximately equal.

$$Mean\ Peptide\ Ratio_{1-n} = \frac{x_1/\bar{x}_1 + x_2/\bar{x}_2 + \ldots + x_n/\bar{x}_n}{n} \tag{1}$$

Precision, shown in S1 Table, was monitored by analysing eight quality control replicates from a CSF pool, which were run with each sample set. The precision and limit of quantification of the method have previously been determined [37]. Given the two sets of samples analysed, the within set variability had coefficients of variation of 1.8–15.8%, depending on peptide. Between sample sets, the coefficients of variation were 2.7–21.0%. For the Mean Peptide Ratio the within set variability coefficients of variation varied between 2.0–13.9% while the between sets variations were 2.1–18.3%.

## Statistical analysis

Statistical analysis was performed with Stata IC 15 software (StataCorp, TX, USA). The distribution of all protein concentrations were tested for normality and found to be non-normally distributed. Natural log-transformation was applied and produced an acceptable distribution for all analytes. Based on their putative involvement in the pathogenesis of HD in the literature, we pre-specified 5 proteins (LAMP1, LAMP2, GM2A, and Cathepsins D and F) and designated them as primary analytes (see S1 Table for full protein list). Differences in demographic and clinical characteristics were examined using ANOVA and $\chi^2$ tests. Age, gender and blood contamination were considered potentially confounding variables, thus their relationship with analyte concentration was examined in a preliminary analysis in controls using independent samples t-tests and Pearson's correlation. Differences across disease stage were tested using general linear models controlling for age. CAG repeat length was also included in the model when assessing differences between premanifest and manifest HD mutation carriers. To test for associations with measures of clinical severity and cognition, Pearson's partial correlation coefficients, bootstrapped with 1000 repetitions, were calculated controlling for age and CAG in all HD gene expansion carriers. Biomarker potential was assessed by controlling relationships first for age, and then for age and CAG. By including both age and CAG as covariates, accurate assessments of associations can be made, independent of known predictors. DBS is a product of age and CAG, as such, the latter two variables were not included as covariates when assessing relationships with DBS.

Principal components analysis (PCA) was employed to reduce the dimensionality of the entire protein dataset. PCA is used to identify the maximum number of uncorrelated principal components that together explain the maximum amount of variance in a data set [64]. We leveraged the Kaiser-Meyer-Olkin measure of sample adequacy and Bartlett's test of sphericity to assess the suitability of our data for PCA. Prior to running the PCA, we controlled each protein for the effect of age using general linear models. When selecting the number of components to use in subsequent analysis, we followed the recommendation to limit this to the smallest number accounting for the most variability in the data [65]. As such, we inspected scree plots and selected components with an eigenvalue of >1. Orthogonal varimax rotation was applied and variables with a loading of >0.3 were deemed significant and used to define the component labels. Participant's original data were then transformed to create a composite score for each principal component. Group differences could then be analysed using this small number of principal components, rather than the large number of original measures. Mirroring the analysis at the level of individual proteins, general linear models and Pearson's partial correlation were used to assess group differences in component scores and the relationships to measures of clinical severity. Age was not included in the models as it had already been controlled for using linear regression in the generation of PC scores.

Significance level was defined as $p < 0.05$ and tests were Bonferroni-corrected for multiple comparisons when required.

A further exploratory analysis was undertaken on the remaining 13 endo-lysosomal proteins using the same hierarchical methodology outlined above.

## Results

### Participant characteristics

Our cohort consisted of 20 healthy controls and 60 HD mutation carriers. The HD gene expansion carriers comprised of 20 premanifest and 40 manifest HD patients. A single premanifest participant was removed due to missing data. There were no significant differences

**Table 1. Demographics and baseline characteristics of each cohort.**

| | Controls (20) | Premanifest (19) | Manifest (40) | ANOVA | Control vs Premanifest | Premanifest vs Manifest |
|---|---|---|---|---|---|---|
| | | | | *p*-value | *p*-value | *p*-value |
| Age (Years) | 50.7 ± 11.0 | 41.8 ± 11.0 | 56.1 ± 9.4 | <0.0001 | 0.008 | <0.0001 |
| Sex (M/F) | 10/10 | 9/10 | 22/18 | NA | NA | NA |
| CAG | N/A | 42.1 ± 1.6 | 42.7 ± 2.1 | NA | NA | 0.22 |
| DBS | N/A | 265.7 ± 63.3 | 395.6 ± 94.6 | NA | NA | <0.0001 |
| TFC | 13 ± 0 | 13 ± 0 | 9.4 ± 2.7 | <0.0001 | 1.00 | <0.0001 |
| TMS | 2.4 ± 2.4 | 2.5 ± 2.6 | 37.5 ± 19.4 | <0.0001 | 0.96 | <0.0001 |
| cUHDRS | 17.4 ± 1.5 | 18.0 ± 1.0 | 10.5 ± 3.6 | <0.0001 | 0.46 | <0.0001 |
| SDMT | 50.9 ± 10.4 | 55.8 ± 9.5 | 27.2 ± 12.6 | <0.0001 | 0.18 | <0.0001 |
| SWR | 100.2 ± 17.4 | 105.5 ± 11.9 | 59.6 ± 23.6 | <0.0001 | 0.40 | <0.0001 |

Intergroup differences were assessed using general linear models and Pearson's chi squared test (Gender). P-values are not adjusted for multiple comparisons. Values displayed are mean ±SD unless otherwise stated. DBS, Disease Burden Score; PRE, Premanifest HD mutation carriers; MAN, manifest HD mutation carriers; CAG, CAG triplet repeat count; cUHDRS, composite Unified Huntington's Disease Rating Scale; SDMT, Symbol Digit Modalities Test; SWR, Stroop Word Reading Test; TFC, Total Functional Capacity; TMS, Total Motor Score; NA, not applicable.

in the gender distribution ($\chi^2$ = 0.34, *p* = 0.84) among the three groups or CAG repeat length among manifest and premanifest HD participants. A significant difference in age was observed, with both healthy controls and manifest HD patients being significantly older than premanifest, because the controls were recruited to span the entire age range of HD mutation carriers. As expected, there were no differences between controls and premanifest individuals in TFC, TMS, cUHDRS, SDMT and SWR, but there were differences between premanifest and manifest HD patients (Table 1).

## Analysis of pre-specified primary analytes

There were no significant differences in protein concentration between genders (LAMP1: Mean Difference (MD) = -0.04, *p* = 0.75; LAMP2: MD = -0.06, *p* = 0.61; GM2A: MD = -0.07, *p* = 0.59; Cathepsin D: MD = -0.07, *p* = 0.49; Cathepsin F: MD = -0.05, *p* = 0.54). CSF haemoglobin concentration, used to evaluate effect of blood contamination, displayed no significant associations with any protein (LAMP1: *r* = 0.16, *p* = 0.49; LAMP2: *r* = 0.09, *p* = 0.70; GM2A: *r* = 0.15, *p* = 0.52; Cathepsin D: *r* = 0.12, *p* = 0.61; Cathepsin F: *r* = -0.13, *p* = 0.59). In addition to significant differences across disease stages, we observed positive trends between CSF protein concentration and age (S1 Fig).

When controlling for age, no significant differences in CSF concentration of LAMP1, LAMP2, GM2A, Cathepsin D or Cathepsin F were observed (group membership main effect: *p* = 0.84; *p* = 0.99; *p* = 0.72; *p* = 0.31; *p* = 0.59, respectively; Fig 1). No significant differences between manifest and premanifest HD patients were observed when also controlling for CAG repeat length (Table 2). Furthermore, we observed no significant differences when grouping together premanifest and manifest HD mutation carriers and comparing with healthy controls (S2 Fig).

Among HD gene expansion carriers, there were no significant correlations between DBS and all measured analytes (Table 3). Furthermore, there were no statistically significant associations between primary analyte concentrations and measures of clinical severity (cUHDRS, TFC, TMS, Fig 2) or cognition (SDMT and SWR, Table 3). Findings remained largely the same when also controlling for CAG repeat length except for LAMP2 which showed a significantly association with TFC (Table 3). Due to LAMP2 demonstrating no significant

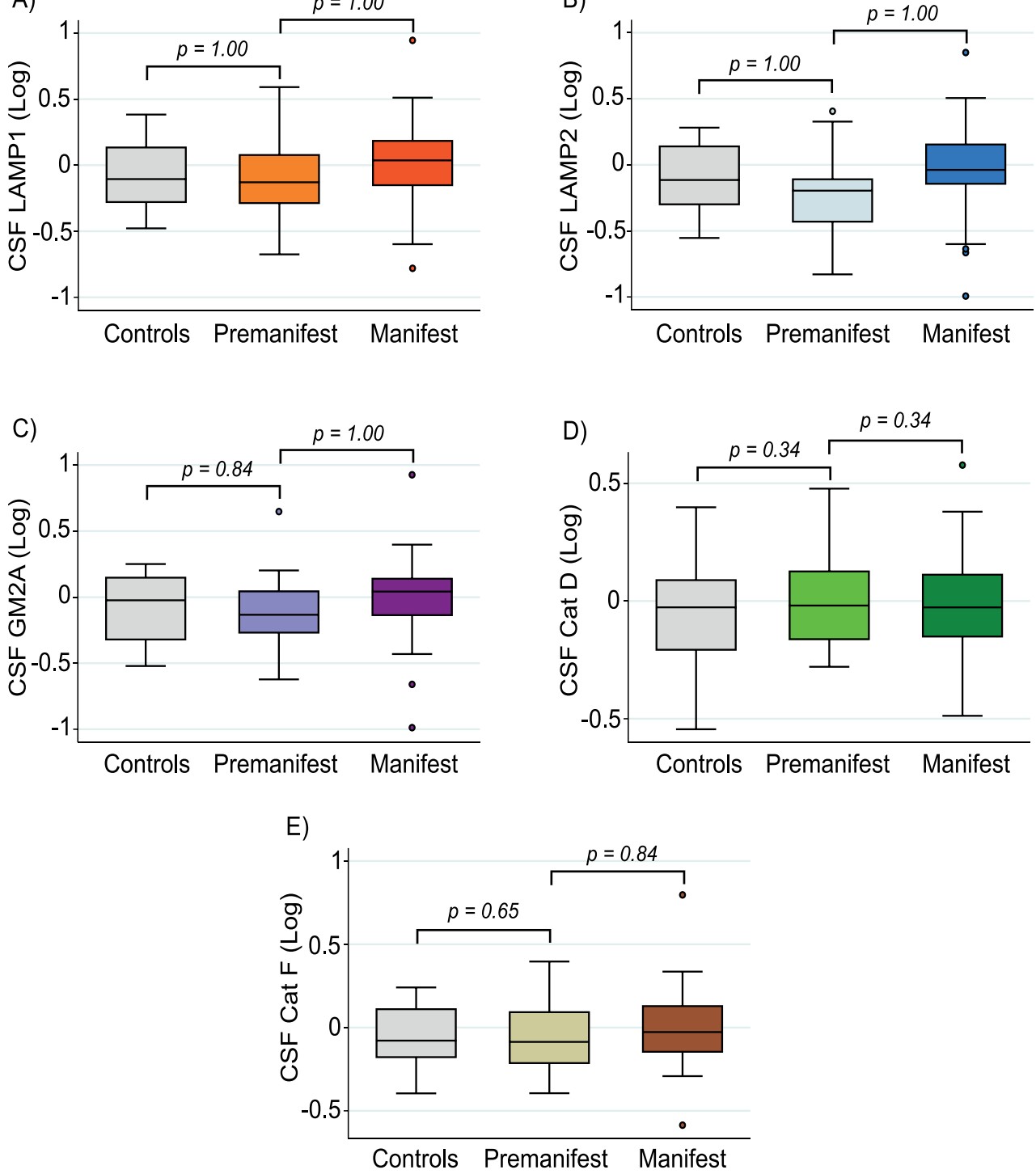

**Fig 1. Comparison of analyte concentration across disease stage.** No significant differences were observed in the concentration of lysosomal (A) LAMP1, (B) LAMP2, (C) GM2A, (D) Cathepsin (Cat) D and (E) Cathepsin (Cat) F between controls, premanifest and manifest HD patients. P-values were Bonferroni-corrected and generated from general linear models controlling for age. Group membership main effects p-values are displayed in text and Table 2. All CSF protein values have been normalized and log-transformed.

**Table 2. Comparison of analyte concentration across disease stage.**

| Endo-Lysosomal Proteins | Adjusted for | ANOVA | Control vs Premanifest | Manifest vs Premanifest |
|---|---|---|---|---|
| | | *p* value | *p* value | *p* value |
| LAMP1 | Age | 0.84 | 1.00 | 1.00 |
| | Age and CAG | NA | NA | 0.70 |
| LAMP2 | Age | 0.99 | 1.00 | 1.00 |
| | Age and CAG | NA | NA | 0.73 |
| GM2A | Age | 0.72 | 0.84 | 1.00 |
| | Age and CAG | NA | NA | 0.64 |
| Cathepsin D | Age | 0.31 | 0.34 | 0.34 |
| | Age and CAG | NA | NA | 0.15 |
| Cathepsin F | Age | 0.60 | 0.64 | 0.86 |
| | Age and CAG | NA | NA | 0.40 |

Differences in analyte concentration across disease stage were assessed using general linear models controlling for effects of age. P-values are Bonferroni-corrected for multiple comparisons when required. CAG was also included in the model when assessing differences between manifest and premanifest HD mutation carriers.

relationship when controlling for age only, it did not meet our criteria for displaying biomarker potential.

## Exploratory principal component analysis

An exploratory principal components analysis was performed on the entire dataset. The Kaiser-Meyer-Olkin measure of sample adequacy was 0.92 and Bartlett's test of sphericity was significant ($\chi^2(153) = 1485$, $p < 0.001$) indicating that PCA was an appropriate means of dimensionality reduction. The first three components (PC1, PC2 and PC3) had eigenvalues of >1 and explained 75% of the variance in the data (59%, 9% and 7%, respectively). A screeplot demonstrated the 'levelling off' of eigenvalues after three components, thus a three-component solution was selected. Composite scores were generated for each of the three components allowing for their use in for subsequent analysis. Based on the protein loadings, the three

**Table 3. Association of analytes and assessed measures in HD mutation carriers.**

| Endo-Lysosomal Proteins | DBS *r* (95% CI) | Adjusted for | cUHDRS *r* (95% CI) | TFC *r* (95% CI) | TMS *r* (95% CI) | SDMT *r* (95% CI) | SWR *r* (95% CI) |
|---|---|---|---|---|---|---|---|
| LAMP1 | 0.27 (-0.05, 0.49) | Age | 0.11 (-0.19, 0.39) | 0.12 (-0.12, 0.37) | -0.08 (-0.34, 0.18) | 0.14 (-0.18, 0.43) | 0.07 (-0.20, 0.34) |
| | | Age and CAG | 0.18 (-0.10, 0.45) | 0.17 (-0.07, 0.41) | -0.14 (-0.36, 0.12) | 0.20 (-0.09, 0.49) | 0.12 (-0.15, 0.39) |
| LAMP2 | 0.32 (-0.02, 0.52) | Age | 0.13 (-0.16, 0.40) | 0.18 (-0.06, 0.41) | -0.10 (-0.34, 0.18) | 0.16 (-0.13, 0.44) | 0.09 (-0.19, 0.34) |
| | | Age and CAG | 0.22 (-0.04, 0.47) | **0.24 (0.01, 0.46)** | -0.16 (-0.40, 0.11) | 0.24 (-0.04, 0.49) | 0.15 (-0.17, 0.45) |
| GM2A | 0.23 (-0.13, 0.45) | Age | 0.13 (-0.14, 0.40) | 0.10 (-0.15, 0.34) | -0.14 (-0.35, 0.10) | 0.17 (-0.15, 0.47) | 0.10 (-0.12, 0.35) |
| | | Age and CAG | 0.15 (-0.12, 0.45) | 0.11 (-0.13, 0.33) | -0.15 (-0.36, 0.12) | 0.19 (-0.09, 0.50) | 0.13 (-0.13, 0.41) |
| Cathepsin D | 0.05 (-0.20, 0.26) | Age | 0.15 (-0.17, 0.40) | 0.11 (-0.17, 0.38) | -0.16 (-0.39, 0.07) | 0.13 (-0.16, 0.41) | 0.13 (-0.14, 0.39) |
| | | Age and CAG | 0.11 (-0.20, 0.39) | 0.07 (-0.21, 0.33) | -0.13 (-0.36, 0.11) | 0.09 (-0.22, 0.43) | 0.09 (-0.19, 0.37) |
| Cathepsin F | 0.25 (-0.07, 0.47) | Age | 0.10 (-0.20, 0.36) | 0.05 (-0.21, 0.30) | -0.13 (-0.35, 0.15) | 0.16 (-0.15, 0.43) | 0.12 (-0.16, 0.36) |
| | | Age and CAG | 0.13 (-0.16, 0.40) | 0.05 (-0.22, 0.31) | -0.14 (-0.37, 0.14) | 0.19 (-0.13, 0.48) | 0.14 (-0.16, 0.40) |

The relationship between protein concentration and Disease Burden Score (DBS) was computed using Pearson's correlation with unadjusted values displayed. Associations with composite Unified Huntington's Disease Rating Scale (cUHDRS), Total Functional Capacity (TFC), Total Motor Score (TMS), Symbol Digit Modalities Test (SDMT), and Stroop Word Reading (SRW) were assessed using Pearson's partial correlation controlling for age, and age and CAG. Correlation coefficients and 95% confidence intervals were computed using bootstrap testing with 1000 repetitions. Results displayed are unadjusted for multiplicity. Bold text indicates p<0.05.

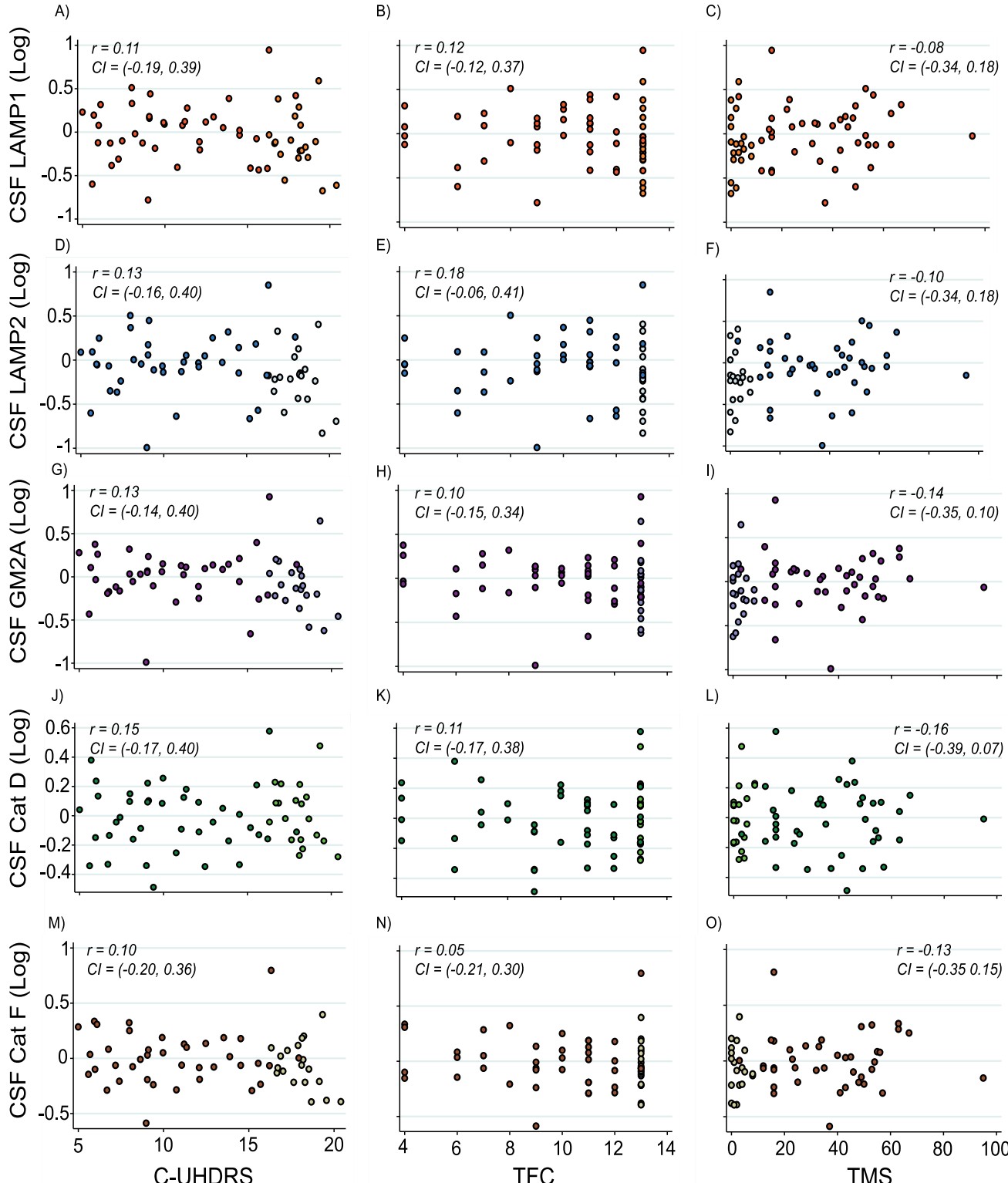

**Fig 2. Correlation between primary analyte concentrations and clinical severity.** Association within HD gene expansion carriers between CSF LAMP1 (A-C), LAMP2 (D-F), GM2A (G-I), Cathepsin (Cat) D (J-L), Cathepsin (Cat) F and composite Unified Huntington's Disease Rating Scale (cUHDRS), Total Functional Capacity (TFC) and Total Motor Score (TMS). Scatter plots show unadjusted values. Correlation coefficients and 95% confidence intervals were generated using Pearson's partial correlation controlling for age and bootstrapped with 1000 repetitions. All CSF protein values have been normalized and log transformed. Lighter coloured data points represent premanifest individuals.

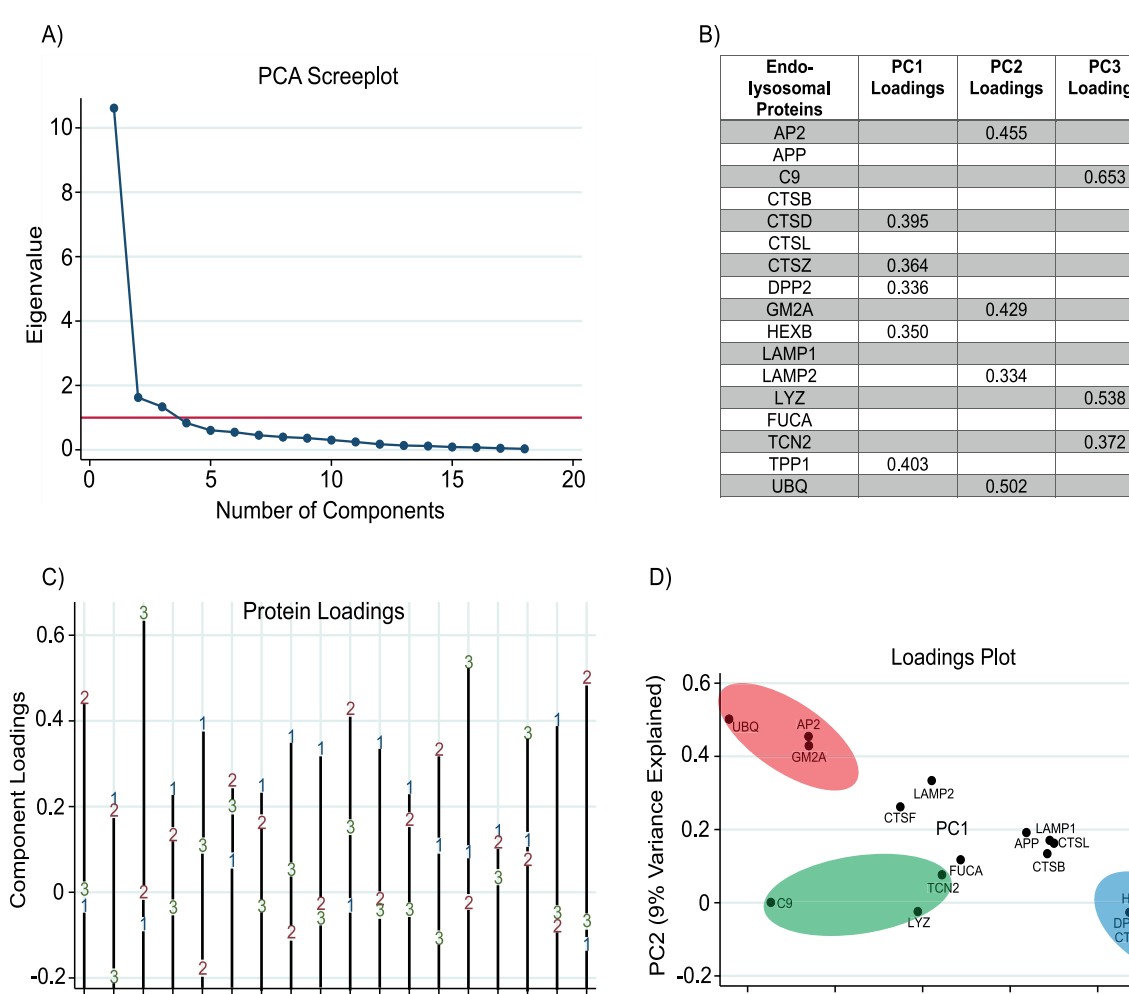

**Fig 3. Screeplot and component loadings following PCA.** (A) Screeplot displays eigenvalues for all components generated. Red line represents an eigenvalue of 1. The first three components have an eigenvalue of >1, thus a three-component solution was adopted. (B) Proteins with loadings of >0.3 were retained and used to define the component labels. All proteins were controlled for age using the residuals from linear regression models. (C) Line graph displaying loadings on the first three components for all proteins included in the PCA. (D) PCA plot demonstrating the clustering of specific proteins into each of the three principal components.

components were deemed to represent lysosomal hydrolases, membrane binding/transfer proteins and innate immune system/peripheral proteins (PC1, PC2 and PC3, respectively) (Fig 3).

The principal component scores for each participant represent a composite that can be used to examine disease-related alterations across all proteins while avoiding multiple comparisons. We found no significant differences in component scores between genders (PC1, $p = 0.65$; PC2, $p = 0.84$; PC3, $p = 0.47$). When comparing across disease stage, we found no significant differences in PC1, PC2 or PC3 (Fig 4). We observed similar findings when CAG was included in the model (S2 Table).

When controlling for age, PC3 demonstrated a significant relationship with TFC only (S3 Table). Composite scores relating to PC1 were not significantly related to any measure of clinical severity or cognition and although PC2 demonstrated a significant relationship with TFC, this relationship was not present when controlling for age only (Fig 5).

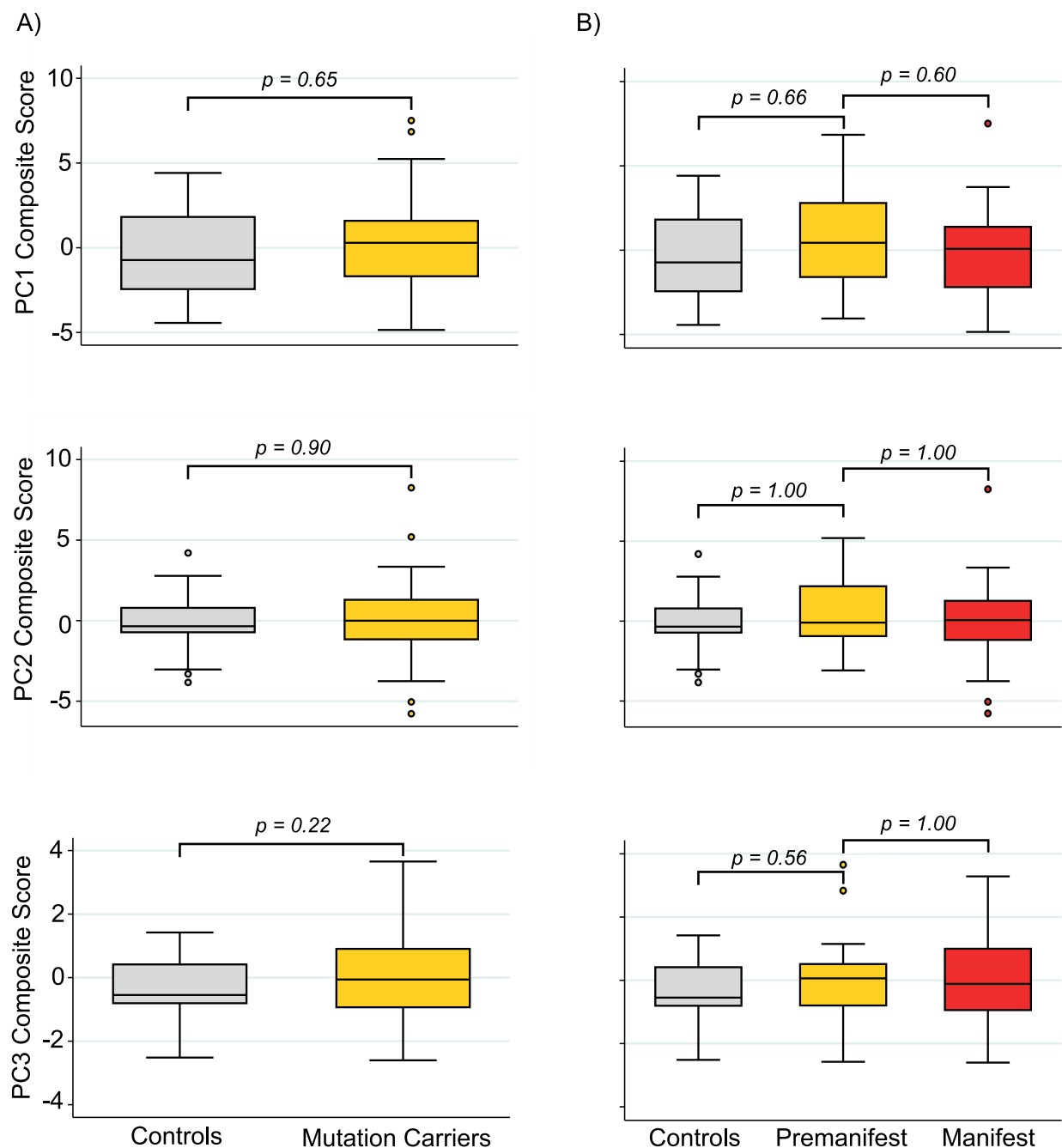

**Fig 4. Group-wise comparison of principal component scores.** No significant differences were observed in Principal component 1 (PC1), 2 (PC2), or 3 (PC3) scores when comparing between healthy controls and GE carriers (A) or across disease stage (B). P-values were Bonferroni-corrected when required and generated from general linear models.

## Exploratory analysis of remaining analytes

Pearson's correlation revealed only C9 and lysozyme C to be significantly associated with age. Nevertheless, we controlled for age in the subsequent analysis of each protein. Lysozyme C also demonstrated a significant gender difference and thus gender was additionally controlled

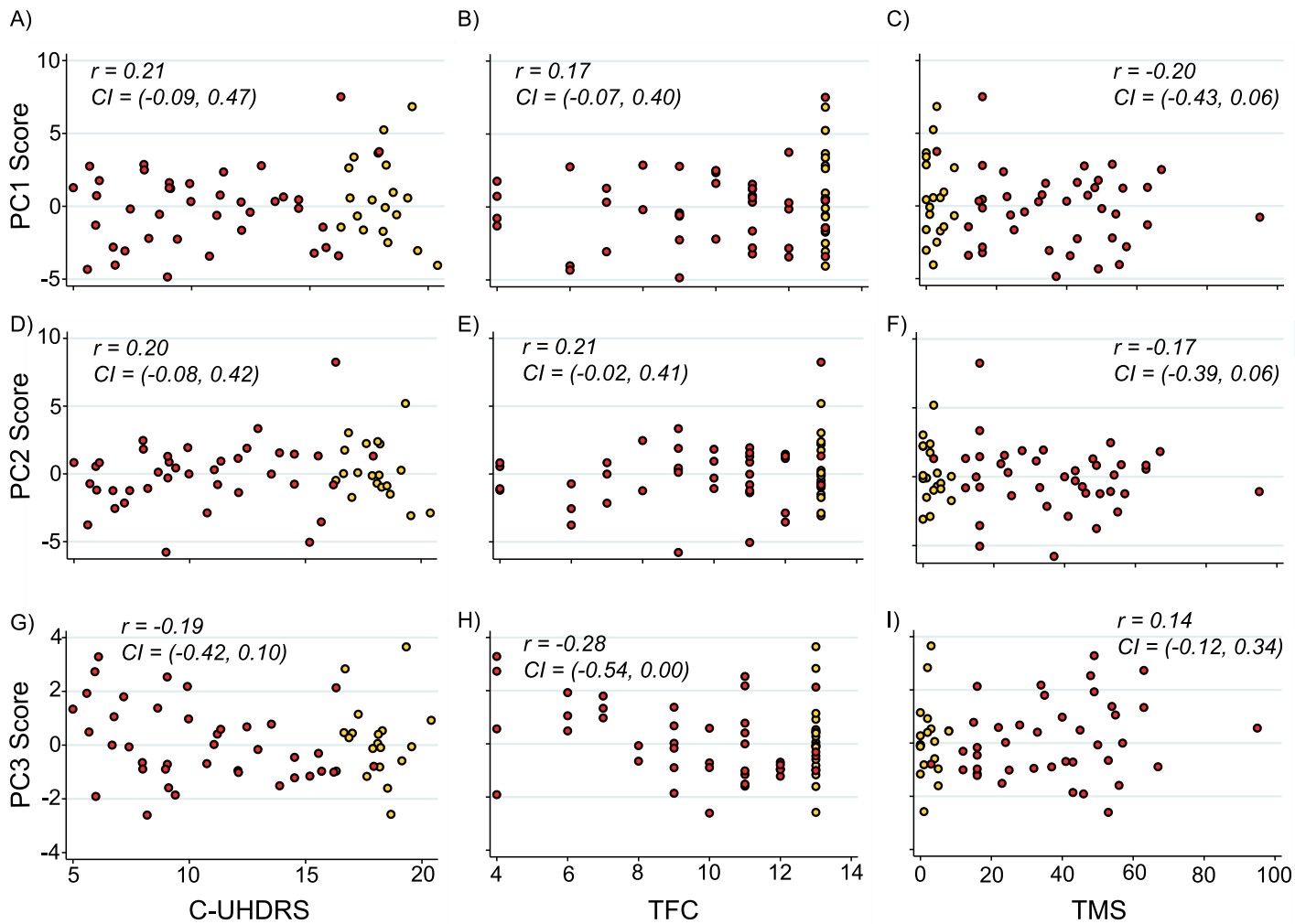

**Fig 5. Correlation between principal component scores and measures of clinical severity.** Association within HD gene expansion carriers between PC1 (A-C), PC2 (D-F), PC3 (G-I), and composite Unified Huntington's Disease Rating Scale (cUHDRS), Total Functional Capacity (TFC) and Total motor score (TMS). Scatter plots show values adjusted for age with correlation coefficients and confidence intervals generated using Pearson's correlation bootstrapped with 1000 repetitions. Red and yellow data points represent manifest and premanifest HD subjects respectively.

for when analysing this protein. No significant associations with haemoglobin concentration were observed (S4 Table).

Despite not showing group-wise alterations (S3 Fig), APP, HEXB, UBQ, Cathepsin B and FUCA were significantly associated with measures of clinical severity within HD mutation carriers when controlling for age. Furthermore, these findings remained significant when additionally controlling for CAG repeat length (Fig 6 and Table 4). Our exploratory analysis of all the remaining endo-lysosomal proteins found no significant differences in analyte concentration across disease stage (S5 Table) or significant relationships with clinical measures, except for C9 and LYZ which displayed significant associations with DBS (S6 Table).

## Discussion

In this cross-sectional study, we successfully quantified 18 endo-lysosomal proteins in high-quality CSF obtained under strictly standardised conditions, from HD mutation carriers and controls, by condensing peptide-level data from 48 peptides quantified using mass

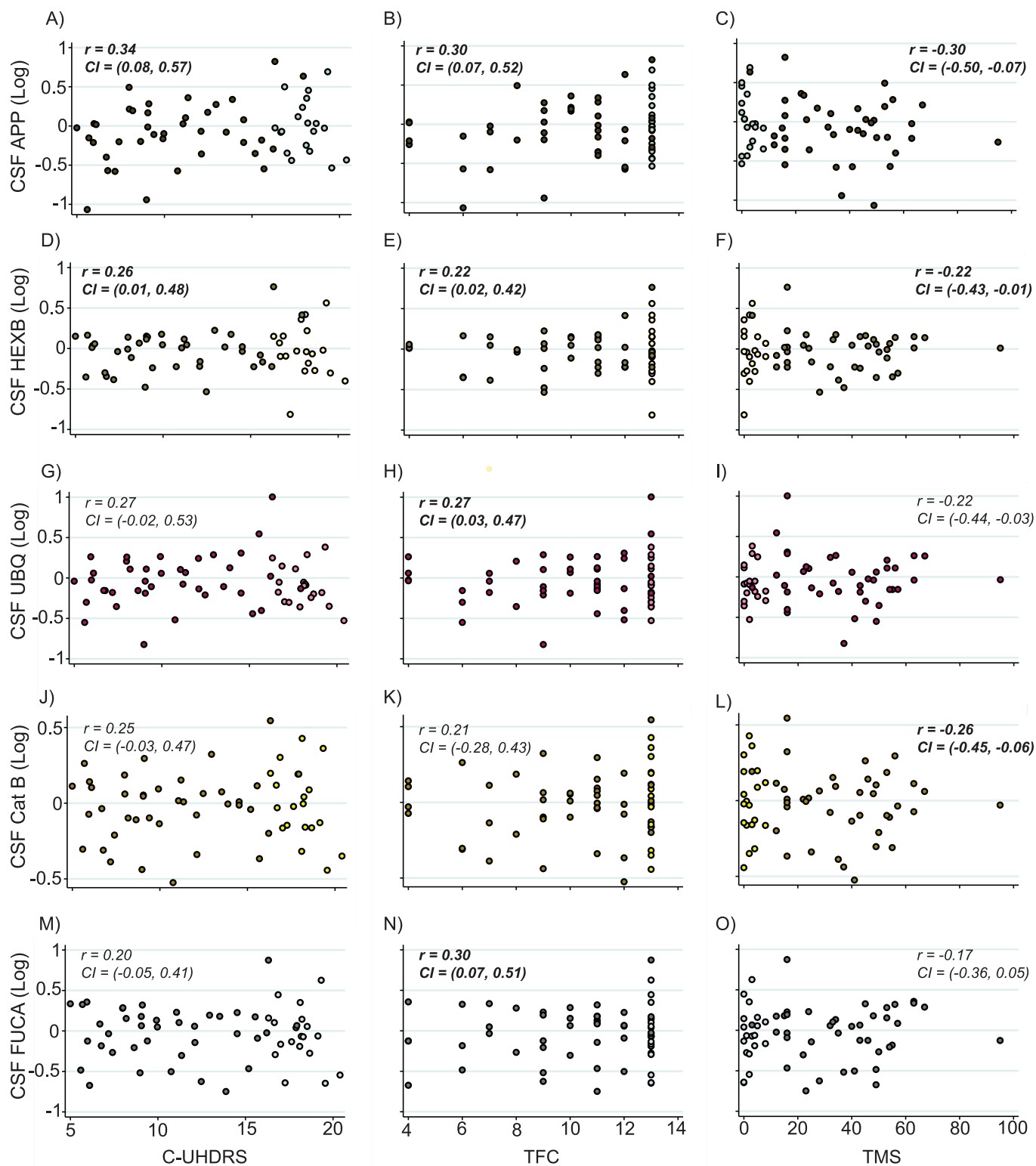

**Fig 6. Significant relationships between exploratory proteins and measures of clinical severity.** Correlation analysis between analyte concentration and composite Unified Huntington's Disease Rating Scale (cUHDRS), Total Functional Capacity (TFC) and Total Motor Score (TMS) revealed significant associations between all three analytes and measures of clinical severity. Scatter plots show unadjusted values. Correlation coefficients and 95% confidence intervals were generated using Pearson's partial correlation controlling for age and bootstrapped with 1000 repetitions. All CSF protein values have been normalized and log transformed. Lighter coloured data points represent premanifest individuals. Bold text indicates significance at p<0.05.

**Table 4. Significant associations between exploratory analytes and assessed measures in HD mutation carriers.**

| Endo-Lysosomal Proteins | DBS r (95% CI) | Adjusted for | cUHDRS r (95% CI) | TFC r (95% CI) | TMS r (95% CI) | SDMT r (95% CI) | SWR r (95% CI) |
|---|---|---|---|---|---|---|---|
| APP | -0.03 (-0.30, 0.24) | Age | **0.34 (0.08 0.57)** | **0.30 (0.07, 0.52)** | **-0.30 (-0.50, -0.07)** | **0.37 (0.09, 0.60)** | **0.32 (0.06, 0.53)** |
| | | Age and CAG | **0.34 (0.06, 0.56)** | **0.27 (0.05, 0.49)** | **-0.27 (-0.46, -0.02)** | **0.36 (0.10, 0.59)** | **0.30 (0.01, 0.53)** |
| HEXB | 0.11 (-0.16, 0.40) | Age | **0.26 (0.01, 0.48)** | **0.22 (0.02, 0.42)** | **-0.22 (-0.43, -0.01)** | **0.27 (0.01, 0.50)** | 0.24 (0.00, 0.47) |
| | | Age and CAG | **0.30 (0.04, 0.56)** | **0.23 (0.01, 0.45)** | **-0.23 (-0.45, -0.02)** | 0.29 (-0.01, 0.59) | 0.26 (-0.04, 0.50) |
| UBQ | 0.13 (-0.17, 0.35) | Age | 0.27 (-0.02, 0.53) | **0.27 (0.03, 0.47)** | -0.22 (-0.44, 0.03) | 0.28 (-0.02, 0.56) | 0.22 (-0.07, 0.47) |
| | | Age and CAG | 0.31 (0.04, 0.55) | **0.28 (0.04, 0.50)** | -0.24 (-0.45, 0.02) | **0.32 (0.03, 0.56)** | 0.24 (-0.09, 0.49) |
| Cathepsin B | 0.12 (-0.17, 0.36) | Age | 0.25 (-0.03, 0.47) | 0.21 (-0.28, 0.43) | **-0.26 (-0.45, -0.06)** | 0.22 (-0.06, 0.46) | 0.22 (-0.05, 0.47) |
| | | Age and CAG | **0.30 (0.02, 0.55)** | 0.23 (-0.02, 0.45) | **-0.30 (-0.54, -0.07)** | 0.25 (-0.03, 0.53) | 0.26 (-0.03, 0.53) |
| FUCA | 0.14 (-0.21, 0.40) | Age | 0.20 (-0.05, 0.41) | **0.30 (0.07, 0.51)** | -0.17 (-0.36, 0.05) | 0.18 (-0.08, 0.42) | 0.15 (-0.10, 0.36) |
| | | Age and CAG | 0.23 (-0.03, 0.47) | **0.32 (0.07, 0.51)** | -0.18 (-0.40, 0.50) | 0.19 (-0.11, 0.50) | 0.16 (-0.12, 0.42) |

The relationships between exploratory analytes and Disease Burden Score (DBS) were assessed using Pearson's correlation with unadjusted values shown. Relationships with, composite Unified Huntington's Disease Rating Scale (cUHDRS), Total Functional Capacity (TFC), Total Motor Score (TMS), Symbol Digit Modalities Test (SDMT), and Stroop Word Reading (SRW) were assessed using Pearson's partial correlation controlling for age, and age and CAG. Correlation coefficients and 95% confidence intervals were computed using bootstrap testing with 1000 repetitions. Results displayed are unadjusted for multiplicity. Bold text indicates significance at $p < 0.05$.

spectrometry. Our pre-specified analysis of the five endo-lysosomal proteins most likely to show relevant HD-related alterations (Cathepsin D, Cathepsin F, GM2A, LAMP1 and LAMP2) found no discernible differences in concentration between HD mutation carriers and controls. Nor, did we observe any significant relationships between the concentrations of these proteins and measurements of clinical severity or cognition. These findings were supported by an exploratory unbiased PCA of the entire dataset which also showed no groupwise differences in three principal components. The findings of our exploratory analysis of the remaining 13 proteins, were also negative for group-wise differences. However, we observed significant negative associations between CSF APP and all measures of clinical severity and cognitive decline within HD mutation carriers, suggesting that APP, and its cleaved product beta-amyloid (Aβ), may be an important avenue to be explored in HD.

Lower levels of CSF APP were associated with worse clinical phenotype and lower cognitive performance. The strongest relationship was observed with cUHDRS score, a powerful measure of clinical progression that predicts corticostriatal atrophy [58]; this relationship, and all others tested, remained significant when controlling for both age and CAG, indicating that there is predictive value independent from well-known predictors of HD progression [60]. APP is a transmembrane protein with multiple physiological functions, including regulating brain iron homeostasis [66]. In HD, mHTT expression has been linked to brain iron accumulation, particularly within neurons [67], potentially exacerbating disease pathology via reactive oxygen species production and oxidative stress [68]. APP is known to facilitate neuronal iron export [66] and has been shown to be decreased in the R6/2 mouse brain [69]. It has been hypothesised that an inadequate APP response to brain iron accumulation may contribute to iron homeostatic dysfunction [70]. The association between reduced CSF APP and clinical worsening in this study provides some support for APP dysfunction in HD and a possible impact on disease progression.

APP is cleaved by β- and γ-secretase to form Aβ peptides [71, 72]. Although we are not measuring Aβ in this study, our findings also raise interesting questions regarding the biomarker potential of CSF Aβ, a biomarker most associated with AD [73], in HD. Reduced CSF Aβ is well described in the AD literature [74–77], likely as a result of increased amyloid

deposition in the brain and reduced clearance into the CSF [78]. The CSF level often demonstrates an inverse relationship with whole brain amyloid load and CSF tau concentration [77, 79, 80]. However, Aβ in CSF has not been studied in HD to our knowledge. Though amyloid deposition is not a typical feature of HD pathology, our APP findings suggest it is possible that Aβ could also represent a novel monitoring or prognostic biomarker in HD.

Similarly, we observed reduced levels of beta-hexosaminidase-β and Cathepsin B tended to predict a more severe clinical phenotype. Cathepsin B is a lysosomal cysteine protease implicated in the pathology of several neurodegenerative diseases, most notably AD [81] in which it has been shown to contribute to increased Aβ load [82], yet also offers potential neuroprotective and anti-amyloidogenic properties [83, 84]. Contrary to previous studies demonstrating increased levels of CSF and plasma Cathepsin B in PD and AD respectively [37, 85], we found Cathepsin B to offer little value as a state biomarker in HD. However, given its significant relationship with TMS and previous work showing reduced mHTT in response to CTSB overexpression [50], it may possess potential for monitoring disease progression.

Together with the co-factor GM2 activator protein (GM2A), beta-hexosaminidase-β is responsible for the degradation of ganglioside GM2 [86]. Mutations in *HEXB*, resulting in reduced levels of the β-subunit and subsequent accumulation of GM2 in neuronal tissue, are the cause of three fatal, neurodegenerative disorders known as the GM2 Gangliosidoses [87]. In this study, we did not observe any differences in beta-hexosaminidase-β across disease stage; however, given its strong association with all three measures of clinical severity, and the reported dysfunction in lipid synthesis, metabolism and catabolism in HD [39, 40], CSF beta-hexosaminidase-β represents an interesting avenue for future research and could help shed light on the role of generalised lysosomal dysfunction in HD pathogenesis.

Furthermore, we observed significant relationships between ubiquitin and complement component C9 and measures of clinical severity. The ubiquitin-proteasome system (UPS) is a key mechanism of intracellular protein clearance, in which misfolded proteins are polyubiquitinated by ligases, thus targeting the substrate for degradation [88, 89]. Previous proteomic work has demonstrated differences in CSF ubiquitin levels between HD patients and controls, whilst also showing a negative relationship with TFC [90]. However, we did not observe any discernible group differences and found lower CSF ubiquitin to be indicative of worsening clinical phenotype. Given these contrary findings and the abundance of literature implicating UPS alterations in the context of Huntington's disease [91–94], further exploration of CSF ubiquitin in HD is required. C9 is a constituent protein of the innate immune system and is highly expressed by astrocytes, microglia and neurons [95–97]. In HD, mHTT activates the complement system resulting in a cascade of neuroinflammatory responses [98]. Neuroinflammation remains a promising area in the field of biomarker research with additional complement components shown to be upregulated in the plasma of HD patients [99] and CSF YKL-40, a microglial marker, showing disease related elevations and the ability to independently predict clinical severity and neuronal death [100]. We found increased levels of C9 and Lysozyme C (LYZ), another cornerstone of innate immunity, to be associated with a higher DBS. This finding was strengthened by our PCA results in which a single component (PC3) correlated negatively with TFC when controlling for age. Interestingly, the protein which loaded highest onto this component was C9, with LYZ also loading highly, thus further supporting the involvement of the innate immune system in HD.

By measuring several peptides per protein, a more accurate approximation of the abundance of the intact protein can be obtained. Our decision to combine the peptides was influenced by our desire to generate an accurate protein-level estimate. However, it should be noted that individual peptides can be derived from different endogenous fragments of the

protein or may belong to different functional domains, therefore there is value in studying individual peptides in future studies.

Our study has some limitations that should be acknowledged. First, the cross-sectional nature of this study means we cannot fully understand how the measured analytes may vary with disease progression; to do this requires longitudinal data collection. Secondly, HD-CSF was principally designed to study manifest HD, so it has a relatively small number of premanifest HD and control subjects. Future studies should recruit larger numbers of subjects within these groups to help improve generalisability of results across the entire disease course. The HDClarity CSF collection initiative [101] represents a large collection of CSF with longitudinal repeat sampling underway. Furthermore, patients with juvenile HD were not recruited in HD-CSF; thus we cannot extend our findings to this sub-population of HD mutation carriers. Finally, all CSF sampling visits were undertaken at the same time of day following an overnight fast; while this minimises the effect of diurnal variation and diet, it may limit the generalisability of our findings.

In conclusion, out of 5 primary and 13 exploratory endo-lysosomal proteins derived from CSF, we could find no alterations in HD patients compared with healthy controls. In our exploratory analyses, we found interesting associations with disease severity for several proteins of potential pathogenic relevance namely HEXB, Cathepsin B, UBQ, C9 and perhaps most notably, APP. These observations link HD severity to several mechanisms, including lipid catabolism deficits, proteostasis network dysfunction, enhanced neuroinflammatory response and dysregulation of iron homeostasis, and suggest a means for beginning to explore these pathways quantitatively in mutation carriers.

Our overall negative groupwise findings in CSF do not exclude a role of lysosomal dysfunction in the pathogenesis of HD; only that major discernible differences in their concentrations could not be observed in the CSF of HD patients. It remains likely that the endo-lysosomal/autophagy system is implicated in the pathology of, and CNS response to, Huntington's disease. However, our work suggests that endo-lysosomal proteins measured in human CSF are unlikely to be state biomarkers in HD but may show promise as tools for exploring pathways of interest and as pharmacodynamic markers for future drug candidates targeting this system.

## Supporting information

**S1 Fig. Correlation analysis between main CSF analytes and age.** Pearson's correlation revealed positive trends between the concentrations of lysosomal (A) LAMP1, (B) LAMP2, (C) GM2A, (D) Cathepsin (Cat) D, (E) Cathepsin (Cat) F and age in healthy controls. All CSF protein values have been normalized and log transformed.
(EPS)

**S2 Fig. Comparison of analyte concentration between gene expansion carriers and controls.** No significant differences in the concentration of lysosomal (A) LAMP1, (B) LAMP2, (C) GM2A, (D) Cathepsin (Cat) D and (E) Cathepsin (Cat) F, was observed between healthy controls and GE carriers. All CSF protein values have been normalized and log transformed. P-values were Bonferroni-corrected and generated from general linear models controlling for age.
(EPS)

**S3 Fig. Comparison of exploratory analyte concentration across disease stage.** We observed no significant differences across disease stage in APP, HEBX, UBQ, Cathepsin (Cat) B and FUCA. P-values were Bonferroni-corrected and generated from general linear models

controlling for age.
(EPS)

**S1 Table. Complete list of lysosomal proteins.** Information pertaining to all CSF endo-lysosomal proteins used in the study.
(PDF)

**S2 Table. Comparison of principal component scores across disease stage.** Differences in scores across disease stage. P-values were Bonferroni-corrected and generated from general linear models. CAG was included in the model when assessing differences between manifest and premanifest HD mutation carriers.
(PDF)

**S3 Table. Association of principal components and assessed measures in HD mutation carriers.** The relationships between principal components 1 (PC1), 2 (PC2) and 3 (PC3) and Disease Burden Score (DBS) were computed using Pearson's correlation with unadjusted values shown. Relationships with composite Unified Huntington's Disease Rating Scale (cUHDRS), Total Functional Capacity (TFC), Total Motor Score (TMS), Symbol Digit Modalities Test (SDMT), and Stroop Word Reading (SWR) were assessed using Pearson's partial correlation controlling for age, and age and CAG. Correlation coefficients and 95% confidence intervals were computed using bootstrap testing with 1000 repetitions. Results shown are unadjusted for multiplicity. Bold text indicates significance at $p < 0.05$.
(PDF)

**S4 Table. Assessments for potential confounding variables in all proteins.** Values are Pearson's r and t-test statistic. Bold indicates significance at the $p < 0.05$ level.
(PDF)

**S5 Table. Exploratory comparison of additional proteins across disease stage.** Differences in analyte concentration across disease stage were Bonferroni-corrected and generated from general linear models controlling for age, or age and CAG. Gender was also included in the model for LYZ. Bold indicates significance at the $p < 0.05$ level.
(PDF)

**S6 Table. Exploratory correlation analysis between endo-lysosomal proteins and measures of clinical severity and cognition.** Associations between analyte concentration and Disease Burden Score (DBS) were assessed using Pearson's correlation with unadjusted values displayed. Associations with composite Unified Huntington's Disease Rating Scale (cUHDRS), Total Functional Capacity (TFC), Total Motor Score (TMS), Symbol Digit Modalities Test (SDMT), and Stroop Word Reading (SWR) were assessed using partial correlation with age, and age and CAG included in the model. For LYZ, the effects of gender were also controlled for. Significant associations are highlighted in bold. Correlation coefficients and confidence intervals were both generated using bootstrapping with 1000 repetitions. Bold indicates significance at the $p < 0.05$ level.
(PDF)

**S1 Dataset. Lysosome dataset.**
(XLSX)

## Acknowledgments

We would like to thank all the participants from the HD community who donated samples and gave their time to take part in this study.

## Author Contributions

**Conceptualization:** Filipe B. Rodrigues, Lauren M. Byrne, Henrik Zetterberg, Edward J. Wild.

**Data curation:** Simon Sjödin, Filipe B. Rodrigues, Lauren M. Byrne, Rosanna Tortelli, Edward J. Wild.

**Formal analysis:** Alexander J. Lowe, Simon Sjödin.

**Funding acquisition:** Henrik Zetterberg, Edward J. Wild.

**Investigation:** Alexander J. Lowe, Simon Sjödin.

**Methodology:** Simon Sjödin, Henrik Zetterberg.

**Project administration:** Kaj Blennow, Edward J. Wild.

**Resources:** Kaj Blennow, Henrik Zetterberg, Edward J. Wild.

**Supervision:** Filipe B. Rodrigues, Lauren M. Byrne, Henrik Zetterberg, Edward J. Wild.

**Validation:** Filipe B. Rodrigues, Edward J. Wild.

**Visualization:** Edward J. Wild.

**Writing – original draft:** Alexander J. Lowe.

**Writing – review & editing:** Simon Sjödin, Filipe B. Rodrigues, Lauren M. Byrne, Kaj Blennow, Rosanna Tortelli, Henrik Zetterberg, Edward J. Wild.

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
