## [Decision Letter · Decision Letter 0]

23 Jun 2020

PONE-D-20-14019

Cerebrospinal fluid endo-lysosomal proteins as potential biomarkers for Huntington’s disease.

PLOS ONE

Dear Dr. Wild,

Thank you for submitting your work to PLOS ONE. Please make the corrections posed by Reviewer #1 so I can render a decision on this manuscript.

Comments to the Author

1. Is the manuscript technically sound, and do the data support the conclusions?

Reviewer #1: Yes

2. Has the statistical analysis been performed appropriately and rigorously? 

Reviewer #1: Yes

3. Have the authors made all data underlying the findings in their manuscript fully available?

Reviewer #1: Yes

4. Is the manuscript presented in an intelligible fashion and written in standard English?

Reviewer #1: Yes

5. Review Comments to the Author

Reviewer #1: Review of “Cerebrospinal fluid endo-lysosomal proteins as potential biomarkers for Huntington’s disease” PLOS ONE 5-26-20 by Lowe et al.

Dysregulated proteostasis and autophagy requiring the endo-lysosomal system have been implicated in neurodegenerative disease pathogenesis. The manuscript under review describes novel, original research determining whether 18 different CSF endo-lysosomal and ubiquitin-proteasome system proteins, including some shown previously to be altered in Alzheimer’s disease (AD) and Parkinson’s disease (PD), might prove to be useful biomarkers for the study of Huntington’s disease (HD) progression. These proteins were detected with parallel reaction monitoring mass spectrometry (PRM-MS) in CSF from Huntington’s disease (HD) mutation carriers and healthy controls and showed no group-wise differences. However, 5 of the proteins including APP, HEXB, UBQ, Cathepsin B and FUCA were significantly associated with measures of clinical severity within HD mutation carriers when controlling for age and CAG repeat length. In particular, negative associations between CSF APP and all measures of clinical severity and cognitive decline within HD mutation carriers were observed, suggesting that APP and Abeta might be useful as HD monitoring or prognostic biomarkers. Increased levels of innate immune system proteins C9 and Lysozyme C were found to be associated with a higher Disease Burden Score (DBS), supporting the involvement of the innate immune system in HD. The conclusions of the study are appropriately supported by the data, and the manuscript is well written. The experimental analyses and statistics were performed to a high technical standard and are described in sufficient detail. The research meets the standards for ethics of experimentation and research integrity and adheres to appropriate reporting guidelines and community standards for data availability.

I have only one concern with the writing of the manuscript in the introduction, starting with line 56 suggesting there are only 2 autophagic mechanisms, macroautophagy and chaperone-mediated autophagy (CMA). Autophagy has been a very fast-moving field over the last 15 years, and the introduction cites some outdated information. Macroautophagy involves the engulfment of cargo into a double-membrane vesicle termed the autophagosome which fuses with vesicles from the endo-lysosomal compartment before becoming an autolysosome, in which its cargo is degraded. Other autophagy processes, which do not involve formation of double membrane autophagosomes, are microautophagy, endosomal microautophagy and CMA. These processes proceed with a direct engulfment of cargo into the endo-lysosomal compartment, either through production of intraluminal vesicles or direct import (Birgisdottir and Johansen, 2020). The Lamp2 protein, including the Lamp-2A splice variant previously only defined in CMA, has now been found to be involved in syntaxin-17-mediated vesicle fusion in macroautophagy (PMID: 27628032), while syntaxin-17 has also been shown to be required for mitochondrial single-membrane vesicle fusion to the endo-lysosomal system in microautophagy (PMID: 27458136). KFERQ-tagged proteins, once thought to be only cleared by CMA have been found to be also degraded by endosomal microautophagy requiring Hsc70 for membrane-fusion events (PMID: 26590345). Recent blurring of the defining lines and machinery between macroautophagy, microautophagy and CMA make understanding the autophagic alterations in PD, AD, polyQ disorders and HD more challenging, especially in light of the fact that the HTT protein itself has been found to be an autophagic scaffold, mutation of which may impair some forms of selective autophagy (PMID: 25385587,PMID: 25686248).

Recent reviews of relevance to the introduction that might be useful to the authors:

PMID: 30149183 Croce and Yamamoto, A Role for Autophagy in Huntington's Disease.

PMID: 31786267 Valionyte et al. 2020: Lowering Mutant Huntingtin Levels and Toxicity: Autophagy–Endolysosome Pathways in Huntington’s Disease.

PMID: 31887286 Djajadikerta et al., 2020: Autophagy Induction as a Therapeutic Strategy for Neurodegenerative Diseases

Birgisdottir A.B., Johansen T. (2020) Autophagy and endocytosis – interconnections and interdependencies. J. Cell Sci 133, jcs228114.

6. PLOS authors have the option to publish the peer review history of their article (what does this mean?). If published, this will include your full peer review and any attached files.

Do you want your identity to be public for this peer review? For information about this choice, including consent withdrawal, please see our Privacy Policy.

Reviewer #1: No

We look forward to receiving your revised manuscript.

Kind regards,

Stephen D. Ginsberg, Ph.D.

Section Editor

PLOS ONE

2.

We note that you have indicated that data from this study are available upon request. PLOS only allows data to be available upon request if there are legal or ethical restrictions on sharing data publicly. For more information on unacceptable data access restrictions, please see http://journals.plos.org/plosone/s/data-availability#loc-unacceptable-data-access-restrictions.

3.

Thank you for stating the following in the Competing Interests section:

I have read the journal's policy and the authors of this manuscript have the following

competing interests: AJL, FBR, LMB, RT, HZ and EJW are University College London

employees. FBR has provided consultancy services to GLG and F. Hoffmann-La

Roche Ltd. EJW reports grants from Medical Research Council, CHDI Foundation, and

F. Hoffmann-La Roche Ltd during the conduct of the study; personal fees from

Hoffman La Roche Ltd, Triplet Therapeutics, PTC Therapeutics, Shire Therapeutics,

Wave Life Sciences, Mitoconix, Takeda, Loqus23. All honoraria for these consultancies

were paid through the offices of UCL Consultants Ltd., a wholly owned subsidiary of

University College London. HZ has served at scientific advisory boards for Denali,

Roche Diagnostics, Wave, Samumed and CogRx, has given lectures in symposia

sponsored by Fujirebio, Alzecure and Biogen, and is a co-founder of Brain Biomarker

Solutions in Gothenburg AB (BBS), which is a part of the GU Ventures Incubator

Program.

---

## [Author Response · Author response to Decision Letter 0]

26 Jun 2020

Reviewer 1

Critique: 

I have only one concern with the writing of the manuscript in the introduction, starting with line 56 suggesting there are only 2 autophagic mechanisms, macroautophagy and chaperone-mediated autophagy (CMA). Autophagy has been a very fast-moving field over the last 15 years, and the introduction cites some outdated information. Macroautophagy involves the engulfment of cargo into a double-membrane vesicle termed the autophagosome which fuses with vesicles from the endo-lysosomal compartment before becoming an autolysosome, in which its cargo is degraded. Other autophagy processes, which do not involve formation of double membrane autophagosomes, are microautophagy, endosomal microautophagy and CMA. These processes proceed with a direct engulfment of cargo into the endo-lysosomal compartment, either through production of intraluminal vesicles or direct import. The Lamp2 protein, including the Lamp-2A splice variant previously only defined in CMA, has now been found to be involved in syntaxin-17-mediated vesicle fusion in macroautophagy, while syntaxin-17 has also been shown to be required for mitochondrial single-membrane vesicle fusion to the endo-lysosomal system in microautophagy. KFERQ-tagged proteins, once thought to be only cleared by CMA have been found to be also degraded by endosomal microautophagy requiring Hsc70 for membrane-fusion events. Recent blurring of the defining lines and machinery between macroautophagy, microautophagy and CMA make understanding the autophagic alterations in PD, AD, polyQ disorders and HD more challenging, especially in light of the fact that the HTT protein itself has been found to be an autophagic scaffold, mutation of which may impair some forms of selective autophagy.

We apologise for the lack of clarity and precision in our original submission and thank the reviewer for this important steer and for taking the time to make such constructive and detailed suggestions for improving the introduction. We have updated and expanded the paragraph in question. Please see below for the changes made, starting at line 56 and ending at line 87 of the resubmitted manuscript:

“….a lysosomal pathway that serves to eliminate toxic substances via three mechanisms, namely microautophagy, macroautophagy and chaperone-mediated autophagy (CMA) [4]. Macroautophagy involves the engulfment of cargo into a double-membrane sequestering vesicle known as an autophagosome. Following fusion with vesicles from the endo-lysosomal compartment, an autolysosome is formed in which the cargo is degraded by lysosomal hydrolases and the resultant macromolecules are released back into the cytosol [5]. Microautophagy and CMA do not involve the formation of an autophagosome, instead using direct import or intraluminal vesicle formation to engulf cargo into the endo-lysosomal compartment [6]. Despite their differences, mechanistic crossovers between the three autophagy pathways have been described. Lysosomal-associated membrane protein-2 splice variant A (LAMP-2A), previously only described in CMA, has been shown to be important for syntaxin-17 mediated vesicle fusion in macroautophagy [7]. Furthermore, syntaxin-17 is pivotal for targeting mitochondrial-derived vesicles to the endo-lysosomal compartment for degradation in microautophagy [8]. 

Macroautophagy plays a pivotal role in the clearance of aggregated proteins via aggrephagy [4,9], whereby aggregates are selectively bound to the autophagosome membrane through the action of adaptor proteins, including p62 and Alfy [10]. Autophagy disruptions have been reported in several neurodegenerative diseases including HD, in which basal autophagy appears to function normally; however, the autophagosomes are devoid of cargo, as recruitment of mHTT to the organelle fails [11–17]. Interestingly, HTT shows structural similarities to three selective autophagy proteins in yeast [18,19] and promotes selective macroautophagy in mammalian cells by mediating the binding of p62 and the autophagy-initiating kinase, UKL1 [20]. As such, it is possible that the polyglutamine expansion in HD may disrupt HTT’s role in selective autophagy [21]; however, studies have shown that autophagic clearance of aggregates can still occur despite overexpression of mHTT in mice and cellular models [22,23]. In light of HTT’s role as an autophagic scaffold protein, the mechanistic crossovers between the three pathways, and their possible contribution to neurodegeneration, we sought to study the alterations and autophagic dysfunction in HD mutation carriers and controls.

---

## [Editor Report · Decision Letter 1]

1 Jul 2020

Cerebrospinal fluid endo-lysosomal proteins as potential biomarkers for Huntington’s disease.

PONE-D-20-14019R1

Dear Dr. Wild,

We’re pleased to inform you that your manuscript has been judged scientifically suitable for publication and will be formally accepted for publication once it meets all outstanding technical requirements.

Kind regards,

Stephen D. Ginsberg, Ph.D.

Section Editor

PLOS ONE

---

## [Editor Report · Acceptance letter]

7 Jul 2020

PONE-D-20-14019R1 

Cerebrospinal fluid endo-lysosomal proteins as potential biomarkers for Huntington’s disease. 

Dear Dr. Wild:

I'm pleased to inform you that your manuscript has been deemed suitable for publication in PLOS ONE. Congratulations! Your manuscript is now with our production department. 

Kind regards, 

on behalf of

Dr. Stephen D. Ginsberg 

Section Editor

PLOS ONE